# Regulatory Network Analysis in Estradiol-Treated Human Endothelial Cells

**DOI:** 10.3390/ijms22158193

**Published:** 2021-07-30

**Authors:** Daniel Pérez-Cremades, Ana B. Paes, Xavier Vidal-Gómez, Ana Mompeón, Carlos Hermenegildo, Susana Novella

**Affiliations:** Department of Physiology, Faculty of Medicine and Dentistry, INCLIVA Biomedical Research Institute, University of Valencia, 46010 Valencia, Spain; daniel.perez@uv.es (D.P.-C.); anapaesmarti@gmail.com (A.B.P.); xavier.vidal@uv.es (X.V.-G.); ana.mompeon@uv.es (A.M.); susana.novella@uv.es (S.N.)

**Keywords:** miRNA, estradiol, endothelial cells, transcription factor

## Abstract

Background/Aims: Estrogen has been reported to have beneficial effects on vascular biology through direct actions on endothelium. Together with transcription factors, miRNAs are the major drivers of gene expression and signaling networks. The objective of this study was to identify a comprehensive regulatory network (miRNA–transcription factor–downstream genes) that controls the transcriptomic changes observed in endothelial cells exposed to estradiol. Methods: miRNA/mRNA interactions were assembled using our previous microarray data of human umbilical vein endothelial cells (HUVEC) treated with 17β-estradiol (E2) (1 nmol/L, 24 h). miRNA–mRNA pairings and their associated canonical pathways were determined using Ingenuity Pathway Analysis software. Transcription factors were identified among the miRNA-regulated genes. Transcription factor downstream target genes were predicted by consensus transcription factor binding sites in the promoter region of E2-regulated genes by using JASPAR and TRANSFAC tools in Enrichr software. Results: miRNA–target pairings were filtered by using differentially expressed miRNAs and mRNAs characterized by a regulatory relationship according to miRNA target prediction databases. The analysis identified 588 miRNA–target interactions between 102 miRNAs and 588 targets. Specifically, 63 upregulated miRNAs interacted with 295 downregulated targets, while 39 downregulated miRNAs were paired with 293 upregulated mRNA targets. Functional characterization of miRNA/mRNA association analysis highlighted hypoxia signaling, integrin, ephrin receptor signaling and regulation of actin-based motility by Rho among the canonical pathways regulated by E2 in HUVEC. Transcription factors and downstream genes analysis revealed eight networks, including those mediated by JUN and REPIN1, which are associated with cadherin binding and cell adhesion molecule binding pathways. Conclusion: This study identifies regulatory networks obtained by integrative microarray analysis and provides additional insights into the way estradiol could regulate endothelial function in human endothelial cells.

## 1. Introduction

The role of 17β-estradiol (E2) in regulating of vascular function, mainly through endothelium-mediated actions [1], could partially account for the sex differences in cardiovascular diseases observed in epidemiological data [2]. 

The effect of E2 on endothelial cell function has been described through direct and/or indirect modulation of gene expression at transcriptional level [3,4]. In addition, regulation through non-coding RNAs has emerged as a key mechanism modulating gene expression profiles. MicroRNAs (miRNAs) are small non-coding RNAs that regulates gene expression of their target messenger RNAs (mRNAs) at the post-transcriptional level via sequence-specific interactions [5,6]. Each miRNA can regulate up to hundreds of targets, and they are predicted to regulate 60% of protein coding genes. 

Specific miRNAs have been identified as crucial modulators of endothelial functions [7], and sex differences in miRNA expression have been reported in different physiological and pathological conditions, suggesting a role for sex hormones in miRNA regulation [8,9]. However, the relationship between estradiol-dependent miRNA expression and cardiovascular function is as yet understudied [10,11]. Our group previously described the role of estrogen receptors (ERs) in regulating of E2-modified miRNAs in endothelial cells [12]. Indeed, both classical nuclear receptors, ERα and ERβ, and membrane-bound ER, G protein-coupled estrogen receptor (GPER), are involved in regulating of E2-sensitive miRNAs [12].

Together with transcription factors, miRNAs are the major drivers of gene expression and signaling networks. Indeed, different studies have demonstrated the role of transcription factors in the protective action that E2 exerts on the vascular function. For example, E2 decreases VCAM1 expression and monocyte adhesion by inhibiting NFkB transcriptional activity in endothelial cells [13], stimulates eNOS expression via activation of the transcription factor Sp1 [14] and inhibits ET-1 gene expression by interfering with the hypoxia-inducible transcription factor (HIF) activity in pulmonary artery endothelial cells [15].

In this study, we performed an integrative analysis of miRNA and mRNA microarray data obtained from human umbilical vein endothelial cells (HUVEC) exposed to physiological concentration of E2. HUVEC are a widely used primary mature endothelial cell model and have been validated in a range of biological processes, including cardiovascular physiology and pathophysiology. This design enables a more in-depth study of specific miRNA/mRNA networks using both predictive and experimental data. Moreover, considering that transcription factors regulate gene expression through binding to transcription factor binding sites in the promoter region of their sensitive genes, we used computational tools that predict target genes of the E2-induced miRNA-regulated transcription factors. Altogether, our study systematically investigated the regulatory mechanism involved in the effect of E2 on endothelial cells, identifying specific miRNA–transcription factor–downstream genes networks. Gene set enrichment analysis was also performed. These findings may provide additional insights into how estradiol regulates endothelial function in human endothelial cells.

## 2. Results

### 2.1. Integrative Analysis of miRNA–mRNA Expression Pairings in E2-Treated HUVEC

An approach combining miRNA and mRNA expression profiles and bioinformatics analysis was arranged to identify the miRNA and mRNA interactions from E2-treated human endothelial cells. Our study integrated data from two distinct datasets: (1) mRNA expression data obtained from E2 (1 nmol/L)-treated endothelial cells using GeneChip Human Genome U133A plus 2.0 Array [16]; (2) miRNA expression profiling previously performed in E2 (1 nmol/L)-treated human endothelial cells using a GeneChip miRNA 4.0 Array expression data [12]. Differential expression analysis of miRNA profile showed 114 differentially expressed miRNA in E2-treated cells compared to control cells (Figure 1A). miR-30b-5p, miR-487a-5p, miR-6734, miR-501-3p and miR-25-5p were the top upregulated miRNAs, while miR-1244, miR-378h, miR-4298, miR-4428 and miR-5588-3p showed the greatest reduction in expression in E2-treated cells compared to control (Figure 1B). Whole transcriptome analysis showed 2204 differentially expressed mRNAs (Figure 1C). The top five up- and downregulated genes based on fold change are represented in Figure 1D. 

Differentially expressed miRNA and mRNA were used to reconstruct global miRNA–mRNA interactions in E2-treated HUVEC. Integrative analysis uses bioinformatics mRNA target prediction and inversely correlated expression obtained from the mRNA microarray test. Figure 2A shows a schematic diagram of the analysis workflow implemented in integrated miRNA–mRNA expression in E2-treated HUVEC and the results obtained. In addition, to adopt a restrictive approach, we further filtered the miRNA–mRNA pairings, selecting only “experimentally observed” and/or “highly predicted” target correlations and discarding those with “moderately predicted” target correlations. After applying these filters, 102 miRNAs were associated with 588 mRNA targets, all of which had opposite expression pairing between miRNA and mRNA levels. Among these, 63 miRNAs were upregulated (with a total of 295 mRNA targets) and 39 miRNAs were downregulated (with a total of 293 mRNA targets). miRNA–mRNA pairings, fold change, confidence filter and target prediction database information are summarized in Appendix A. Figure 2B represents the fold change of the top miRNA-predicted targets (showing only the top five targets based on fold change). 

### 2.2. Enrichment Pathway Analysis of miRNA–mRNA Expression Pairings in E2-Treated HUVEC

miRNA–mRNA pair interactions with opposite expression obtained were used to perform pathway analysis in IPA software. Ingenuity Canonical Pathway database identified 73 significantly enriched pathways (Appendix A). Enrichment analysis showed that miRNA targets were associated with different biological processes related to endothelial function. The top ten canonical pathways based on statistical significance, enrichment ratio and number of hits per group are shown in Figure 3A. Among these, E2-related miRNA targets are associated with hypoxia signaling, integrin-linked kinase (ILK) signaling, integrin signaling and the regulation of actin-based motility by RhoGDI signaling. In addition, Figure 3B represents differentially expressed genes included in the top five most significant signaling pathways presented as a chord plot.

### 2.3. Identification of miRNA-Regulated Transcription Factors in E2-Treated HUVEC

For a better understanding of the transcriptomic changes associated with E2 effect on endothelial cells, we identified transcription factors among the E2-regulated miRNAs targets. In total, 13 differently expressed miRNA targets were identified as transcription factors using EnrichR software (Appendix A). miRNA and the fold change of their transcription factor target are represented in Figure 4A. Transcription factors JUN and RREB1 were identified among the targets of E2-upregulated miRNAs miR-30b-5p and miR-25-5p, respectively. E2-induced miR-6808 targets three differentially expressed transcription factors: CBX5, PPARA and ATF1. Finally, transcription factor PLAG1 was identified as a predicted target of miR-1296-5p and miR-26a-5p. Two transcription factors, TEAD2 and REPIN1, were identified as predicted targets of the E2-downregulated miR-4685-5p. Furthermore, we wanted to know the regulatory gene sets of these transcription factors. Different tools have been developed to identify transcription factor binding sites (TFBS) within the promoter region of downstream genes regulated by a specific transcription factor. We used both TRANSFAC and JASPAR tools from EnrichR website to search for TFBS of the miRNA-regulated transcription factors within the promoter region of the E2-regulated genes. In total, 712 genes were identified as downstream genes regulated by E2-sensitive transcription factors (Appendix A). Among them, JUN and REPIN1 were the transcription factors with greatest number of predicted downstream genes. Specifically, 297 E2-regulated genes were associated with JUN, while REPIN1 TFBS was predicted in the promoter region of 209 E2-regulated genes. PPARA was associated with 21 differentially expressed genes found in E2-treated cells. Finally, RREB1, CBX5 and PLAG1 were associated with five downstream genes each one. We also used transcription factor and related downstream genes to perform gene set enrichment analysis (Figure 4B), finding that ATF1, JUN, PPARA, REPIN1 and TEAD2 gene sets were associated with different pathways from GO, KEGG and Reactome databases. No enriched pathways were found for RREB1, CBX5 and PLAG1 gene sets due to the low number of predicted downstream genes. The results show that the most significant enriched pathways based on *p* value and number of genes involved were the “cadherin binding” (*p* = 1.32 × 10^−5^, 18 hits) pathway within the JUN gene set and the “cell adhesion molecule binding” pathway (*p* = 9.39 × 10^−5^, 17 hits) that is associated with REPIN1-related downstream genes. Among downstream predicted genes, FSCN1, ADD1, DDX3X and PTPRB are commonly regulated by JUN and REPIN1, as shown in the network visualization of these two pathways (Figure 4C). In addition, the ATF1 gene set was found to be involved in different interferon-related pathways and stress-triggers responses, while PPARA was associated with “steroid hormone receptor activity” and “nuclear receptor activity”. Finally, enrichment analysis showed that, in addition to REPIN1 and JUN, TEAD2 was also involved in the “focal adhesion” category (*p* = 3.81 × 10^−4^, 3.50 × 10^−3^ and 8.36 × 10^−4^, respectively).

## 3. Discussion

In the present work, we used a comprehensive regulatory network analysis to unravel the transcriptomic changes observed in endothelial cells exposed to E2. First, miRNA and mRNA expression profiles were systematically analyzed in order to find miRNA–mRNA pairing interactions. Along with miRNA target prediction algorithms, joint analysis of miRNA and mRNA expression profile data obtained using the same culture conditions (HUVEC exposed to 1 nmol/L E2 for 24 h) increased the process-specificity of predicted miRNA–mRNA interactions. We further selected the transcription factors among the miRNA target genes regulated by E2 and identified their predicted downstream target gene sets. The use of gene set enrichment analysis tools revealed new potential regulatory mechanisms associated with the effects of E2 on the endothelium. 

Integrated analysis revealed miRNA–targets pairings showing inversely correlated expression in E2-treated cells. Only those experimentally observed or highly predicted were selected for further analysis. For example, miR-30b-5p, the most upregulated miRNA in endothelial cells after E2 treatment, was predicted to target 24 differentially expressed genes whose expression was decreased in E2-treated cells (Appendix A), some of which were previously reported as targets of miR-30 family members, including PPP3CA [17], RAB32 [18] and MAP4K4 [19] (Figure 2B). PPP3CA, which encodes a catalytic subunit of calcineurin, is regulated by miR-30 family members in podocytes and cardiomyocytes [17]; in this regard, estrogen action on PPP3CA expression has already been reported, since estradiol exposition decreases PPP3CA expression and reduces cardiomyocyte hypertrophy [20], which could explain the sex differences observed in cardiac hypertrophy [21]. In addition, RAB32 is a Rab GTPase involved in vesicular trafficking and autophagosome formation [18]; MAP3K2 has been implicated in the regulation of apoptosis regulation [22]; MAP4K4 regulates endothelial cell activation, since MAP4K4 silencing inhibits cell adhesion molecule expression [23]. In this regard, it is noteworthy that E2 plays a role in autophagy [24] and triggers anti-inflammatory and anti-apoptotic mechanisms in the endothelium [3].

Enrichment analysis of differentially expressed transcripts regulated by E2 in endothelial cells indicated a dysregulation of pathways related to: (1) hypoxia signaling; (2) ILK signaling; (3) integrin signaling. Hypoxia signaling pathway was the most significantly enriched pathway, which includes different miRNA-regulated genes such as HSP90AB1, P4HB and several E2 ubiquitin activating enzymes (UBE2s). HSP90AB1 and P4HB, both upregulated in E2-treated cells, are considered HIF1-associated chaperons. Additionally, most UBE2s enzymes were downregulated in cells exposed to E2, which has been proposed as a HIF1 regulatory mechanism [25]. These findings suggest that E2 could regulate HIF1 by regulating associated chaperons and inhibiting its proteosomal degradation. Along these lines, George et al. reported that estrogen can molecularly mimic hypoxia by activating HIF1, which increases VEGF expression to promote angiogenesis [26].

Ephrin receptor signaling was also highlighted among the top canonical pathways regulated by E2 and is involved in the regulation of critical aspects of endothelial function by controlling cell-to-cell contact [27]. Indeed, loss of ephrin molecules in the heart disrupted the structural integrity of cardiac endothelium leading to a cardiomyocyte hypertrophy [28].

Estrogens induce endothelial cell migration and angiogenic activity in the endothelium, in which the interactions between endothelial cells and the surrounding extracellular matrix play a crucial role [29]. As a critical connection between intracellular structural components and the extracellular matrix, integrins and integrin-linked kinase (ILK) signaling enable control of cytoskeletal organization and cell motility. In this regard, estrogens also increase expression of integrins, which play an important role in mediating endothelial cell attachment and migration [30]. Accordingly, “regulation of actin-based motility by Rho” and “RhoGDI signaling” were also highlighted by our analysis. In this regard, Rho GTPases family signaling has been implicated in regulation of E2-mediated endothelial cell migration, since E2-treated HUVEC showed activated RhoA activity, which in turn increased actin cytoskeleton formation and cell migration [31]. Moreover, estrogens are involved in an enhanced re-endothelialization after vascular injury [32], partly mediated by mobilization of endothelial progenitor cells to the site of injury [33]. This process is regulated by the CXCR4 pathway, another canonical pathway highlighted in our analysis. 

In addition to the effect of estrogens on integrin signaling described above, adherens junctions are also necessary for actin remodeling and endothelial migration induced by E2 in endothelial cells [34,35]. In this sense, functional enrichment analysis highlighted “cadherin binding” pathway as among the most significant regulated pathways within the JUN-associated gene set (Figure 4B,C). Indeed, E2-dependent regulation of adherens junctions has been linked to Src kinases activity, which regulates JUN [35]. JUN dimers or JUN–FOS heterodimers constitute the AP1 complex, which plays an important role in cell migration and proliferation. ERs interact with AP1 to regulate expression of genes with AP1 sites in their promoter region, and ER has been reported to inhibit stress-induced c-Jun activity [36]. However, ERs can act as transcription activators or repressors when regulating AP1-dependent genes [37].

REPIN1 (also known as AP4) was found among the transcription factors regulated by E2-responsive miRNAs and functions as both activator and repressor of gene transcription [38,39]. REPIN1 activity has been reported in transcriptional repression of cell cycle inhibitor p21, regulating cell cycle progression [40] or inducing the epithelial–mesenchymal transition (EMT) and migration in cancer cells [41]. Indeed, genome-wide analyses of AP4-overexpressing cells showed dysregulation of several EMT genes, including induction of SNAIL, FN1, CDH2, VIM, TGF-β and several MMPs, while CDH1, CHD3, KRT19, PKP3 and different claudin and integrin genes were suppressed [41]. Concurrent with this, we found CDH5 (cadherin 5), ITGA5 (integrin a5) and PKP4 (plakophilin 4, a desmosomal plaque component involved in regulating junctional and cadherin function) among the REPIN1 downstream genes (Figure 4C). Endothelial–mesenchymal transition (EndoMT) is a process in which endothelial cells lose cell–cell interactions, acquiring migratory properties, and regulated by molecular mechanisms similar to those of EMT [42]. Accordingly, estrogens have been implicated in regulation of angiogenesis regulation in adenomyosis using EndoMT-related signaling pathways [43].

This study has limitations. First, no fold-change cut-off was applied when using pathway analysis tools, regardless of the *p* value. Second, there was no functional validation of the findings of this study. Gain- and loss-of-function experiments using miRNA mimics and inhibitors and their impact in target gene expression and biological function will provide insights into the endothelial regulation by E2. Third, other competitive endogenous RNAs (ceRNAs), such as pseudogenes and long non-coding RNAs, that might antagonize the miRNA expression, were not analyzed in this study.

## 4. Material and Methods

### 4.1. Cell Culture and Experimental Design

Human umbilical vein endothelial cells (HUVEC) were obtained from human umbilical cords from healthy women at the Department of Obstetrics and Gynecology of the Hospital Clínico Universitario of Valencia, as previously described [44]. Briefly, endothelial cells were detached using 0.1 mg/mL collagenase (Life Technologies, Carlsbad, CA, USA) incubation for 15 min at 37 °C. Isolated HUVEC were cultured in Medium 199 (Sigma-Aldrich, Madrid, Spain) supplemented with 20% fetal bovine serum (GIBCO, Life Technologies), endothelial cell growth supplement (Sigma-Aldrich), heparin sodium salt (Sigma-Aldrich) and antibiotics (GIBCO, Life Technologies). Cells were cultured at 37 °C and 5% CO_2_, and cultured media were changed every 2–3 days.

Cells were identified as endothelial by their characteristic cobblestone morphology and the presence of von Willebrand factor (vWF) by immunofluorescence using a specific antibody (ab6994, Abcam, Cambridge, UK). More than 95% of HUVEC cultures were positive for vWF. HUVEC from passages 3 to 4 were seeded onto gelatin-coated 6-well plates. When they reached confluence, culture medium was exchanged for a phenol red-free Medium 199 (Sigma-Aldrich) supplemented with 20% charcoal/dextran-treated fetal bovine serum (GIBCO, Life Technologies) to avoid any steroid activity, and it was maintained for 24 h. Next, cells were exposed for 24 h with 1 nmol/L of E2 (Sigma-Aldrich) or vehicle (0.1% ethanol, control).

Our research using HUVEC conformed to the principles outlined in the Declaration of Helsinki, was approved by the Clinical Research Ethics Committee of INCLIVA–Hospital Clínico Universitario of Valencia and we obtained written informed consent from all umbilical cord donors.

### 4.2. Integrative Analysis of miRNA–mRNA Expression Profile 

Global changes in miRNA and mRNA expression of HUVEC treated with 1 nmol/L E2 for 24 h were previously described by our group using microarrays technology [12,16]. Integrative analysis of miRNA–mRNA interactions was determined based on database target predictions and differential expression data. 

Differentially expressed miRNA and mRNA datasets were selected to perform integrative analysis using Ingenuity Pathways Analysis (IPA) software (Ingenuity Systems, Redwood City, CA, USA; fall release 2015). This software uses computational algorithms to identify miRNA targets as well as putative cellular networks related to predicted targets. The IPA microRNA Target Filter tool enables prioritization of experimentally validated and predicted mRNA targets according to three different miRNA target prediction programs (TargetScan, miRecords and Ingenuity Knowledge Base) and the available database of experimentally supported miRNA targets, TarBase. Inverse expression between E2-dependent miRNAs and its predicted mRNA targets was determined among differentially expressed mRNAs. A confidence filter was used to select both experimentally observed and highly predicted target correlations. 

### 4.3. Pathway Enrichment Analysis

Differentially expressed genes (DEGs) were identified as *p* value < 0.05. DEGs were subjected to gene set enrichment analyses using MetaCore™ (v21.2, Clarivate Analytics, Boston, MA, USA) and Ingenuity Pathway Analysis (IPA, Qiagen, fall release 2015) software. Enrichment analysis for functional ontologies (Process Networks) and analysis using network building tools were performed in MetaCore™. The MetaCore default setting of false discovery rate (FDR) < 0.05 was used as the threshold for significance in enrichment analysis. For canonical pathway analysis, IPA (fall release Dec 2019) was used. The pathway activity (Z score) was computed to determine whether canonical pathway activity was increased or decreased on the basis of differentially expressed genes in the datasets. The significant values for canonical pathways were calculated by Fisher exact test. Pathway enrichment analysis of specific gene sets was performed using R package clusterProfile [45], which uses different pathway repositories, including Gene Ontology (GO), which annotates genes using ontology [46]; Kyoto Encyclopedia of Genes and Genomes (KEGG) pathway repository consisting of curated reference pathway maps, which are then mapped to genes within different organisms based on orthologous associations [47]; Reactome database, whose annotations are manually curated from literature by expert biologists [48]. Categories were selected based on *p* value (*p* < 0.1), and the visualization of pathway enrichment analysis were performed as a dot plot using (ggplot2 package) and circos plot (circlize package) in R program [49].

### 4.4. Identification of Transcription Factors—Target Networks

The regulatory network between miRNA-regulated transcription factors and their targets were predicted using JASPAR/TRANSFAC databases within Enrichr software (maayanlab.cloud/Enrichr/, accessed on 19 April 2021). Differentially expressed (*p* < 0.05) transcription factor targets were selected.

## 5. Conclusions

This paper describes an integrative analysis of miRNA and mRNA expression profiles conducted in human endothelial cells exposed to E2. Our results show specific miRNA–transcription factor–downstream genes networks and their related biological processes, pointing to the role of E2-regulated miRNAs in transcriptomic changes in endothelial functions. These findings make an open contribution to future research and may provide new insight into the mechanisms by which estradiol can contribute to sex differences in cardiovascular diseases. 

## Figures and Tables

**Figure 1 ijms-22-08193-f001:**
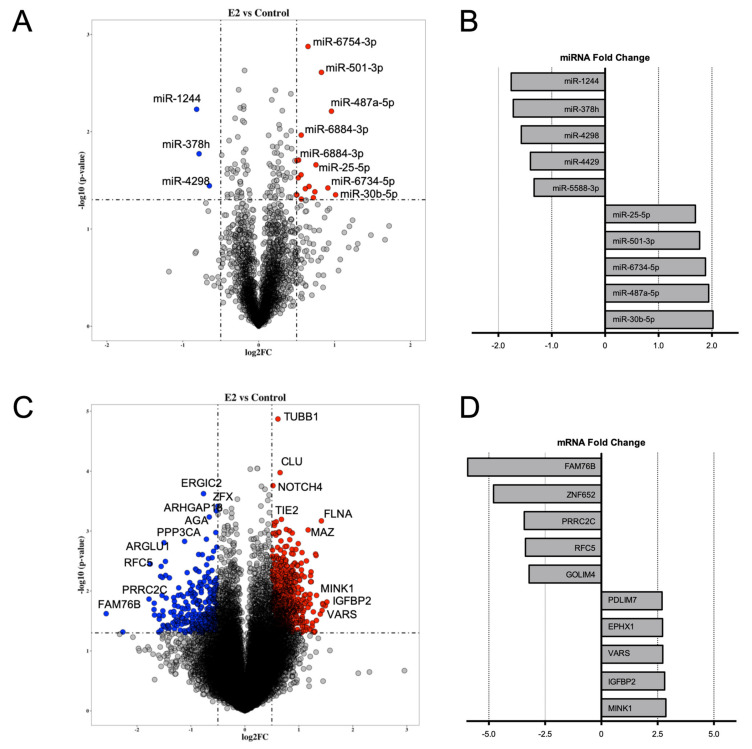
Distribution of differentially expressed miRNAs and mRNAs obtained using microarray of estradiol–exposed HUVEC. HUVEC were treated with vehicle (0.1% ethanol) or estradiol (1 nmol/L) for 24 h and non-coding RNA were determined using microarray technology. Volcano plot highlighting differentially expressed miRNAs (**A**) and mRNAs (**C**) between E2-treated and control samples (*p* value < 0.05 and log2 fold change > 0.5). Upregulated hits on upper right quadrant (red dots) and downregulated hits on upper left quadrant (blue dots). (**B**) Top five up- and downregulated miRNAs (**B**) and mRNAs (**D**) are shown based on fold change.

**Figure 2 ijms-22-08193-f002:**
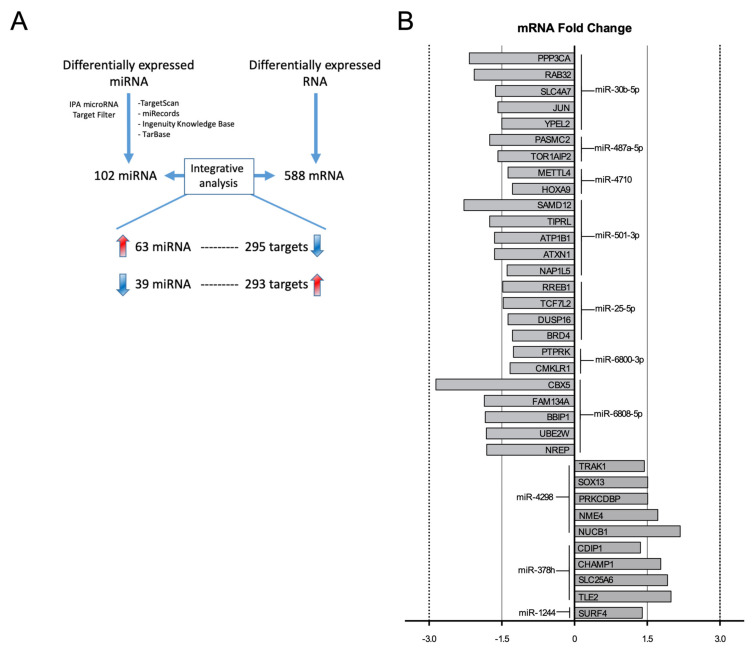
Workflow of integrative analysis of miRNA–mRNA pairings in E2–treated HUVEC. (**A**) Workflow analysis. miRNA–target pairings were filtered by using differentially expressed miRNA and mRNA characterized by a regulatory relationship according to miRNA target prediction databases (TargetScan, miRecords, Ingenuity Knowledge Base and TarBase) using Ingenuity Pathway Analysis software. Confidence filter was also implemented, selecting only experimentally observed or highly predicted target correlations. The analysis identified 588 miRNA–target interactions between 102 miRNAs and 588 targets. Specifically, 63 upregulated miRNAs interact with 295 downregulated targets, while 39 downregulated miRNAs pairs with 293 upregulated mRNA targets. Red and blue arrows indicate miRNAs or targets that are up- or downregulated, respectively. (**B**) miRNA–target interactions of the top miRNAs. Fold changes of the top five predicted targets are shown.

**Figure 3 ijms-22-08193-f003:**
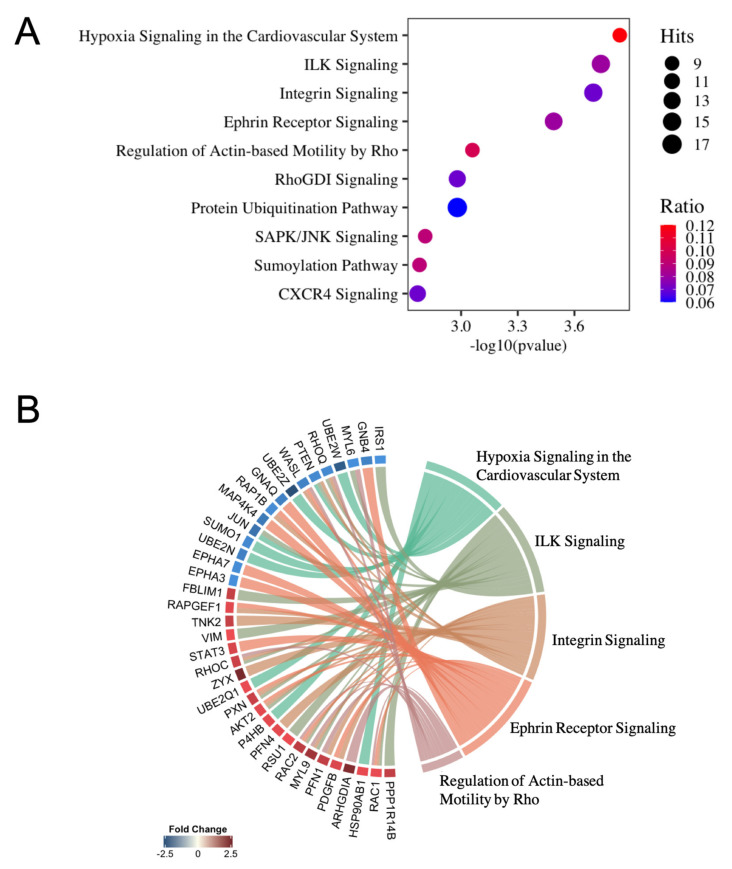
Canonical pathway study of inversely correlated miRNA–mRNA pairs using IPA software. (**A**) High predicted and experimentally observed pairings among inversely correlated miRNA–mRNA interactions were used to determine canonical pathways. *p*-value is determined by the probability that each biological function assigned to the network is due to chance alone. Hits represent the number of differentially expressed genes regulated per network. Ratio represents the amount of differentially expressed genes obtained in a given pathway divided by total number of genes in a specific canonical pathway. (**B**) Chord plot representing associations between differentially expressed genes and their canonical pathways.

**Figure 4 ijms-22-08193-f004:**
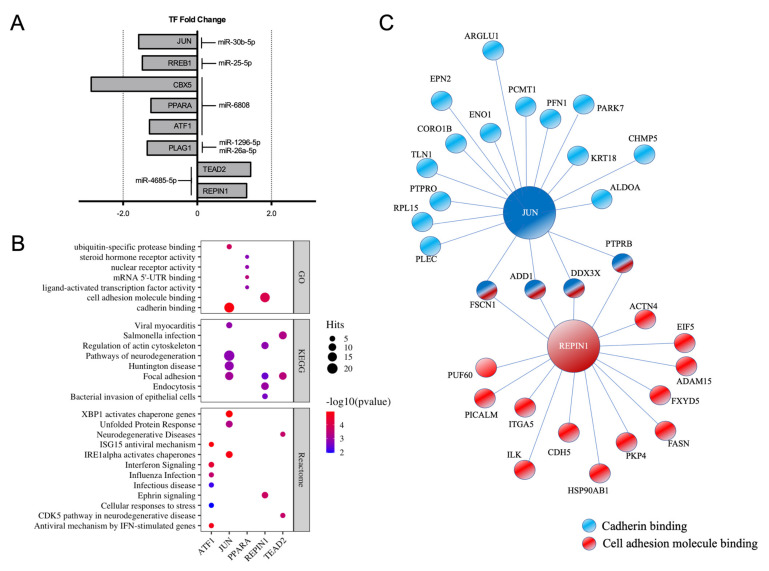
Regulatory networks induced by estradiol in HUVEC. Transcription factors targets and down-stream genes were identified among the predicted targets using TRANSFAC-JASPAR databases from Enrichr tools. (**A**) miRNA-targeted transcription factors are depicted. Fold changes of the transcription factors targets are shown. (**B**) Dot plot of GO, KEGG and Reactome pathway enrichment for the identified transcription factors. Dot size and color represent number of hits within each pathway and *p* value, respectively. (**C**) Network visualization of enriched pathways: “cadherin binding” and “cell adhesion molecule binding”.

## Data Availability

All results obtained in this study are available in the Appendix A.

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
