# Peer review of "Regulatory Network Analysis in Estradiol-Treated Human Endothelial Cells"

_ijms, 2021, doi:10.3390/ijms22158193_

Round 1
Reviewer 1 Report
In this review, Pérez-Cremades et al. establish a great number of new pathways involved in the role of estrogen at vascular level. Through bioinformatics, authors perform an integrative analysis of miRNA and mRNA expression profile in HUVEC exposed to 17β-estradiol. Moreover, they show the transcription factors affected by deregulated miRNA by 17β-estradiol, which open new research possibilities.
In general, the study is well conducted and with recent literature. However, there are some minor concerns I would appreciate to be discussed.
- Regarding Figure 4, could authors explain the biological reason for which 17β-estradiol modulate different transcription factors (JUN, REPIN1) through opposed expression of miR-30b-5p and miR-4685-5p? This effect leads to an increase and a decrease of the same transcription factors (FSCN1, ADD1, DDX3X and PTPRB) that apparently does not make sense.
- Also, although in the 4th position, Ephrin Receptor Signaling pathway appears to be regulated by estradiol. As this pathway has been proposed to mediate cell-to-cell communication, it could be extremely important in the crosstalk between endothelial cells and, for example, cardiomyocytes.
Although the text can be read and understood correctly, a revision of the English style would probably improve the quality of the manuscript.
Please, define E2 as 17β-estradiol the first time it is mentioned in the abstract.
Author Response
In this review, Pérez-Cremades et al. establish a great number of new pathways involved in the role of estrogen at vascular level. Through bioinformatics, authors perform an integrative analysis of miRNA and mRNA expression profile in HUVEC exposed to 17β-estradiol. Moreover, they show the transcription factors affected by deregulated miRNA by 17β-estradiol, which open new research possibilities.
In general, the study is well conducted and with recent literature. However, there are some minor concerns I would appreciate to be discussed.
Regarding Figure 4, could authors explain the biological reason for which 17β-estradiol modulate different transcription factors (JUN, REPIN1) through opposed expression of miR-30b-5p and miR-4685-5p? This effect leads to an increase and a decrease of the same transcription factors (FSCN1, ADD1, DDX3X and PTPRB) that apparently does not make sense.
Thank you for the comment. In Figure 4 we identified transcription factors among the E2-regulated miRNA targets. Therefore, up-regulated miRNAs are associated with down-regulated transcription factors (i.e. miR-30b-5p/JUN), and down-regulated miRNAs are associated with up-regulated transcription factors (i.e. miR-miR-4685-5p/REPIN1).
On the other hand, Figure 4C represents the gene set network from two enriched pathways: “Cadherin binding” (depicted in blue) and “Cell adhesion molecule binding” (depicted in red). 4 genes were found in both highlighted pathways, and they are depicted in both red and blue.
However, all predicted downstream genes were selected regardless of their directionality. In that sense, although transcription factors are frequently classified as activators or repressors, their gene regulatory activity is controlled by a combination of factors, including the expression of co-activators and co-repressors, where context-dependency specifies how transcription factors modulates the expression of each specific gene (Genes Dev. 2006; 20: 1405-1428).
Also, although in the 4th position, Ephrin Receptor Signaling pathway appears to be regulated by estradiol. As this pathway has been proposed to mediate cell-to-cell communication, it could be extremely important in the crosstalk between endothelial cells and, for example, cardiomyocytes.
Thanks to the reviewer for this comment. We think this is very interesting and we add some sentences in the Discussion addressing the role of Ephrin pathway in endothelial cell.
“Ephrin receptor signaling was also highlighted among the top canonical pathways regulated by E2, and is involved in the regulation of critical aspects of endothelial function by controlling cell-to-cell contact (27). Indeed, loss of ephrin molecules in the heart disrupted the structural integrity of cardiac endothelium leading to a cardiomyocyte hypertrophy (28).”
Although the text can be read and understood correctly, a revision of the English style would probably improve the quality of the manuscript.
Following your suggestion, the manuscript has been revised by an English language reviser.
Please, define E2 as 17β-estradiol the first time it is mentioned in the abstract.
We have added “17β-estradiol” in the abstract as suggested.
Reviewer 2 Report
The work from Cremades and colleagues goes deeper into the previous work by (Vidal-Gómez et al, Cell Physiol Biochem 2018, 45, 1878-1892), where the regulation of miRNAs expression in HUVEC after estrogen exposure was analysed. Here a new insight into miRNAs-mRNA interaction upon estrogen stimulation is provided. The paper is interesting and overall very clear and well-written and structured. However I would like to highlight some minor points.
- The authors claim in the Introduction that miRNA can regulate gene expression by suppressing the expression of target mRNA (line 42). However, it is now known that miRNAs can both repression and stimulate gene expression (please, see Vasudevan S,Wiley Interdisciplinary Reviews: RNA, vol. 3, no. 3, pp. 311–330, 2012). This must be corrected.
- A pending scientific question is how miRNA concentration, besides just expression, relates to target mRNA suppression. It has been suggested that only highly expressed miRNAs are able to mediate negative post-transcriptional gene regulatory effects. More specifically miRNAs expressed below ~100 copies per cell had little regulatory capacity (Brown BD, et al. Biotechnol. 2007;25:1457–1467). Have the authors selected the relevant miRNAS according not only due to significance but other criteraia such as with a cut-off for copy number or minimum fold-increase?. This should be mentioned and discussed. Otherwise this limitation should be stated in the paper.
- In figure 4B, the authors must be aware that KEGG maps are merely composites of pathways in many organisms -do not identify what specific pathways elucidated in what organisms. Could a more comprehensive analysis be carried out for the KEGG map?. What is the relevance of e.g. Salmonella infection or neurodegeneration in the context of endothelial cells stimulated with estrogen?.
- Have the authors analysed other competitive endogenous RNAs (ceRNAs) of the selected miRNAs; e.g. transcripts with multiple MREs such as pseudogenes, lncRNAs, that might compete or antagonise with the proposed differentially expressed miRNAs?. If not, this limitation should be stated in the paper.
- It has been proposed that the real impact of a functional miRNA/mRNA target pair must be validated by fulfilling four main criteria (Elton & Yalowich, EXCLI J. 2015; 14: 758–790). Mainly:
Showing that the predicted miRNA and mRNA target gene are co-expressed.
Demonstrating a direct interaction of a given miRNA to a specific MRE harbored within the target mRNA.
Gain- and loss-of-function experiments utilizing miRNA mimics and inhibitors must inversely regulate target protein expression.
miRNA-mediated regulation of target gene expression (gain- and loss-of-function) should equate to altered biological function.
The 2 first validations seem to be fulfilled but the last 2 points remain unclear. Could the authors please elaborate on this?. Are there other publications supporting their findings, or have they silenced the miRNAs themselves. Otherwise this limitation should be stated.
Finally, minor typos spelling mistakes should be checked (e.g. line 13 of the abstract).
“...a comprehensive regulatory networks (miRNA-transcription factor- downstream genes) that controls the transcriptomic changes observed in endothelial cells exposed to estradiol.” Singular instead of plural is missing.
Author Response
The work from Cremades and colleagues goes deeper into the previous work by (Vidal-Gómez et al, Cell Physiol Biochem 2018, 45, 1878-1892), where the regulation of miRNAs expression in HUVEC after estrogen exposure was analysed. Here a new insight into miRNAs-mRNA interaction upon estrogen stimulation is provided. The paper is interesting and overall very clear and well-written and structured. However I would like to highlight some minor points.
The authors claim in the Introduction that miRNA can regulate gene expression by suppressing the expression of target mRNA (line 42). However, it is now known that miRNAs can both repression and stimulate gene expression (please, see Vasudevan S,Wiley Interdisciplinary Reviews: RNA, vol. 3, no. 3, pp. 311–330, 2012). This must be corrected.
Thanks to the reviewer for this comment. We have corrected this in Introduction and added the reference.
A pending scientific question is how miRNA concentration, besides just expression, relates to target mRNA suppression. It has been suggested that only highly expressed miRNAs are able to mediate negative post-transcriptional gene regulatory effects. More specifically miRNAs expressed below ~100 copies per cell had little regulatory capacity (Brown BD, et al. Biotechnol. 2007;25:1457–1467). Have the authors selected the relevant miRNAS according not only due to significance but other criteraia such as with a cut-off for copy number or minimum fold-increase?. This should be mentioned and discussed. Otherwise this limitation should be stated in the paper.
We thank the reviewer for this comment. We agree with the reviewer that set up a cut-off of minimum reads, in the case of miRNA-sequencing experiments, or a fold-change threshold it is important to obtain more robust results. However, we decided not to apply any fold-change cut-off in this study, regardless of the p value, providing all significant findings in the supplemental files that can be used as an exploratory resource for future studies.
Anyway, we include this aspect in the last paragraph of the Discussion.
In figure 4B, the authors must be aware that KEGG maps are merely composites of pathways in many organisms -do not identify what specific pathways elucidated in what organisms. Could a more comprehensive analysis be carried out for the KEGG map?. What is the relevance of e.g. Salmonella infection or neurodegeneration in the context of endothelial cells stimulated with estrogen?.
We agree with reviewer’s comment. Although pathway analysis tools have been broadly used in transcriptomic research, there are biases and false positives since genes that are highly ranked in the gene list may lead to enrichment of different pathways. In other words, unrelated pathways to the experiment can also be significantly enriched due to crosstalk genes between pathways (as stated in PMID: 23934932, PMID: 29218878).
In figure 4B we used the R-based tool ClusterProfile to perform gene set enrichment analysis with transcription factor downstream gene sets. We chose to combine data from three pathway repositories that were available in this package, which gave us a broader insight into the related pathways. The three databases are among the largest and most widely used pathway repositories and they employ different approaches to curating and compiling pathway information. For example, GO annotates genes using ontology (PMID: 10802651); KEGG pathway repository consists of curated reference pathway maps, which are then mapped to genes within different organisms based on orthologous associations (PMID: 9287494); and Reactome database annotations are manually curated from literature by expert biologists (PMID: 21067998).
Have the authors analysed other competitive endogenous RNAs (ceRNAs) of the selected miRNAs; e.g. transcripts with multiple MREs such as pseudogenes, lncRNAs, that might compete or antagonise with the proposed differentially expressed miRNAs?. If not, this limitation should be stated in the paper.
We did not analyse other ceRNAs that could be modulating miRNAs. We have included this limitation in Discussion section.
It has been proposed that the real impact of a functional miRNA/mRNA target pair must be validated by fulfilling four main criteria (Elton & Yalowich, EXCLI J. 2015; 14: 758–790). Mainly:
- Showing that the predicted miRNA and mRNA target gene are co-expressed.
- Demonstrating a direct interaction of a given miRNA to a specific MRE harbored within the target mRNA.
- Gain- and loss-of-function experiments utilizing miRNA mimics and inhibitors must inversely regulate target protein expression.
- miRNA-mediated regulation of target gene expression (gain- and loss-of-function) should equate to altered biological function.
The 2 first validations seem to be fulfilled but the last 2 points remain unclear. Could the authors please elaborate on this?. Are there other publications supporting their findings, or have they silenced the miRNAs themselves. Otherwise this limitation should be stated.
We thanks the reviewer for the comment and we are in agreement with that. There was no functional validation of the findings of this study. Gain- and loss-of-function experiments using miRNA mimics and inhibitors and their impact in target gene expression and biological function will provide insights into the endothelial regulation by E2. Indeed, our next efforts will be focused on validating the data obtained in this study to better understand the effects of E2 on endothelial cell.
Following reviewer suggestion, we add this limitation in the last paragraph of the Discussion section.
Finally, minor typos spelling mistakes should be checked (e.g. line 13 of the abstract).
“...a comprehensive regulatory networks (miRNA-transcription factor- downstream genes) that controls the transcriptomic changes observed in endothelial cells exposed to estradiol.” Singular instead of plural is missing.
We thank the reviewer correction. We have proceeded accordingly. Furthermore, the new version of the manuscript has been revised by a native English speaker.
Reviewer 3 Report
In this study, the author used HUVEC as a model to evaluate the miRNA-transcription factor- downstream genes that control the transcriptomic changes in estradiol treated endothelial cells.
The biological roles of estradiol in endothelial cells is worthy to be investigated, the study design and methods are appropriated. Only some questions are suggested to clearly descript.
- Why use HUVEC as a model of endothelial cells, I suggest to introduce it in the part of introduction and discuss other experimental endothelial cells model.
- If there are some different characteristics between HUVEC and other endothelial cells, I suggest changing the topic …”HUMAN ENDOTHELIAL CELLS” to HUVEC.
- The rational of the dose of E2 (1 nmol/l) and the time periods of 24 hours that treated in HUVEC are suggested to state.
4.The network analysis could provide as a screening tool to realize the estradiol effect, I suggest the author describe the limitation of the results, and give some comments on how to combine with the functional study to confirm the correction of the network.
Author Response
In this study, the author used HUVEC as a model to evaluate the miRNA-transcription factor- downstream genes that control the transcriptomic changes in estradiol treated endothelial cells.
The biological roles of estradiol in endothelial cells is worthy to be investigated, the study design and methods are appropriated. Only some questions are suggested to clearly descript.
- Why use HUVEC as a model of endothelial cells, I suggest to introduce it in the part of introduction and discuss other experimental endothelial cells model.
We thank the reviewer for this comment. HUVECs are a widely used endothelial cell model because of their human origin and the high isolation success rate compared to other primary mouse endothelial cells. In addition, HUVEC have been validated as an excellent primary mature endothelial cell model to study a broad range of biological mechanisms where endothelium is involved, including inflammation, cardiovascular diseases, cancer, and regenerative medicine.
Following reviewer’s suggestion, the rationale to use HUVEC as a model of endothelial cells has been incorporated in Introduction section.
- If there are some different characteristics between HUVEC and other endothelial cells, I suggest changing the topic …”HUMAN ENDOTHELIAL CELLS” to HUVEC.
We have carefully considered your suggestion. However, we consider the use of the acronym HUVEC in the title should be explained, resulting in a very large topic, and we maintain the topic as in the original version. Moreover, the rationale to use HUVEC in the study has been addressed, as stated before.
- The rational of the dose of E2 (1 nmol/l) and the time periods of 24 hours that treated in HUVEC are suggested to state.
Thanks to the reviewer for the comment. We used 1 nmol/l E2 as it is a physiological concentration in humans. In addition, previous studies from our group using a range of concentrations and different timepoints showed that the treatment of endothelial cells in vitro with 1 nmol/l E2 during 24h regulates important endothelial pathways such as the production of prostacyclin (PMID: 20110403), nitric oxide (PMID: 22982060), and angiotensin (PMID: 26562171).
- The network analysis could provide as a screening tool to realize the estradiol effect, I suggest the author describe the limitation of the results, and give some comments on how to combine with the functional study to confirm the correction of the network.
We thanks the reviewer for the comment. There was no functional validation of the findings of this study. Our next efforts will be focused on validating the data obtained in this study to better understand the effects of E2 on endothelial cell. Anyway, we add this limitation in the last paragraph of the Discussion section following reviewer’s suggestion.